# A Hypothetical New Challenging Use for Indocyanine Green Fluorescence during Laparoscopic Appendectomy: A Mini-Series of Our Experience and Literary Review

**DOI:** 10.3390/jcm12165173

**Published:** 2023-08-08

**Authors:** Noemi Zorzetti, Augusto Lauro, Manuela Cuoghi, Marco La Gatta, Ignazio R. Marino, Salvatore Sorrenti, Vito D’Andrea, Andrea Mingoli, Giuseppe Giovanni Navarra

**Affiliations:** 1Department of General Surgery, “A. Costa” Hospital—Alto Reno Terme, 40046 Bologna, Italymarco.lagatta@ausl.bologna.it (M.L.G.); giuseppe.navarra@ausl.bologna.it (G.G.N.); 2Department of Surgical Sciences, La Sapienza University, 00186 Rome, Italy; augusto.lauro@uniroma1.it (A.L.); salvatore.sorrenti@uniroma1.it (S.S.); vito.dandrea@uniroma1.it (V.D.); andrea.mingoli@uniroma1.it (A.M.); 3Sidney Kimmel Medical College, Thomas Jefferson University, Philadelphia, PA 19107, USA; ignazio.marino@jefferson.edu

**Keywords:** acute appendicitis, indocyanine green, laparoscopic appendectomy, fluorescence, new technologies

## Abstract

Laparoscopic appendectomy (LA) is a well-standardized surgical procedure, but there are still controversies about the different devices to be used for the appendiceal stump closure and the related postoperative complications. Indocyanine green (ICG) fluorescence angiography (FA) has already been shown to be helpful in elective and emergency surgery, providing intraoperative information on tissue and organ perfusion, thus guiding the surgical decisions and the technical strategies. According to these two aspects, we report a mini-series of the first five patients affected by gangrenous and flegmonous acute appendicitis intraoperatively evaluated with ICG-FA during LA. The patients were admitted to the Emergency Department with an usual range of symptoms for acute appendicitis. The patients were successfully managed by fully LA with the help of a new hypothetical challenging use of ICG-FA.

## 1. Introduction

Acute appendicitis (AA) is the most common abdominal surgical emergency in the world, and its diagnosis is based on history, physical evaluation, laboratory tests, and imaging [1]. Although increasing evidence suggests that broad-spectrum antibiotics or combination therapy successfully treat uncomplicated acute appendicitis in approximately 70% of patients, LA remains the most common treatment [1]. Specific imaging findings on computed tomography (CT) such as appendicolith, mass effect, and a dilated appendix greater than 13 mm are associated with a higher risk of treatment failure (≈40%) of an antibiotics-first approach; so, in these cases, surgical management should be recommended. Non-surgical management, in selected cases, has an acceptable initial success rate, but almost 38% of therapeutic failure occurs in the first 12 months after the first episode, and patients should be carefully informed. Either appendectomy or antibiotics can be considered first-line therapy [1] in patients without high-risk CT findings.

AA is classified as uncomplicated or complicated: uncomplicated appendicitis is defined by acute appendicitis without clinical or radiographic signs of perforation (inflammatory mass, phlegmon, or abscess), while a complicated one is defined by appendiceal rupture with subsequent abscess or phlegmon formation [2]. Gangrenous appendicitis, even without perforation, is considered subsidiary to having suffered bacterial translocation, and in general terms, the literature considers it a complicated appendicitis.

Indocyanine green (ICG) is a low-risk dye used in medicine and surgery, with multiple applications providing “real-time” images by combining white light (WL) images with fluorescence (near infrared, NIR), which can be used for the identification of anatomical structures and the evaluation of tissues vascularization [3]. 

In the following sequence of patients, taking into account both the grade of appendicitis, the vascularized base, and the ICG background, “an idea was born”: using indocyanine green fluorescence angiography (ICG-FA) in order to evaluate if the addition of this dye could add some data to improve the surgical management of this common urgent surgery worldwide.

## 2. Mini-Series and Evolution

Five patients were admitted to the Emergency Department (ED) and were included in this mini-series according to the diagnosis of acute appendicitis.

The socio-demographic characteristics and laboratory tests are collected in Table 1, while the radiological features are collected in Figure 1.

After the evaluation of a general surgeon, all the patients were admitted, and an emergency surgical procedure was recommended. Written informed consent was obtained and antibiotic prophylaxis was given. Laparoscopy was initiated by placing a supraumbilical 12 mm Hasson trocar using an open technique. After pneumoperitoneum induction, two other trocars were placed in the left flank and suprapubic position (10 mm and 5 mm, respectively), with identification of the appendix, cut, and coagulation of the mesoappendix. 

Usually, the choice for the device to manage the base of the appendiceal stump is made by the operating surgeon, evaluating the appendicular base.

## 3. Materials and Methods

We performed a search on PubMed based on the following key words: acute appendicitis, laparoscopic appendectomy, and indocyanine green fluorescence angiography. To the best of our knowledge, we did not find previous publications regarding this specific application for ICG.

Our hypothesis is to explire if ICG-FA during LA, especially for complicated AA, can provide more data about the vascularization of appendicular base and stump, suggesting that, in selected cases, the use of an endo-stapler could be advisable in order to reduce post-operative complications and, first of all, intra-abdominal abscesses (IAA).

All patients were >18 years old, had no allergy to Iodine, and were not in therapy for hyperthyroidism. Written provided informed consent for the use of their data and for the off-label administration of ICG were collected. 

The fluorescence is detected using specific cameras that transmit the signal to a monitor; the technical characteristics of the camera used for this study are the NIR/ICG visualization modes with 4K-3D imaging chain.

For all the patients, the first aim of our study was the evaluation of the appendicular bases with ICG-FA before proceeding with resection (Figure 2). Patients received endovenous (ev) administration of ICG according to this dilution: one vial of 25 mg was diluted in 10 cc of saline solution, and 5 cc were given at the first peak between 20 and 40 s. The first infusion was administered to evaluate good blood flow perfusion of the appendix bases once they were completely visualized. Considering that the appendix bases were not gangrenous, such as the cecum, and thanks to this confirmation and integrity after 40–60 s from the infusion of ICG-FA, endo-loop was used in all procedures. The appendicular stump was then dissected 3–5 mm away from cecum.

After the resection, in *P3*, *P4*, and *P5*, the stump was assessed using ICG-FA because of the significant gangrenous grade of these appendicitis (Figure 3). The second aim was therefore to evaluate the appendicular stump with ICG-FA, administrating the remnant 5 cc of solution and waiting for the peak at 20–40 s. 

In our patients, good blood tissue perfusion of the stump and of the cecum was confirmed (Figure 4), no futher indication for endostapler was given, and the surgical specimen was then removed in an endo-bag using the 12 mm trocar.

At the end of the procedure, peritoneal lavage was conducted, and an abdominal drain was left in the pelvis. Antibiotic therapy was recommended according to the recent literature [4]. All patients were discharged in good clinical condition. 

## 4. Discussion and Literary Review

ICG is a historical dye and low-risk molecular compound for patients used in different fields of medicine and both elective and emergency surgery that has been around since the mid-1950s [3,5,6,7]. 

Combined in real-time WL images with fluorescence NIR, ICG provides intraoperative information on blood perfusion and tissue vitality, guiding the successive surgical strategy [8,9]. 

For several years, in colorectal surgery, ICG showed feasibility and usefulness in the intraoperative assessment of vascular anastomotic perfusion, leading, if necessary, to changing the site of resection and/or anastomosis, thus reducing the anastomotic leak (AL) rate [10,11]. Consequently, the outcome and patients‘ safety improve, also decreasing the need for surgical reintervention for anastomotic leaks [12,13]. 

Also, in emergency intestinal surgery (for example, acute mesenteric ischemia, strangulated ileus, and incarcerated hernia), ICG-FA demonstrated a role in preventing undue intestinal resection or its entity, facilitating the identification of the intestinal ischemic zone, detecting the well-perfused intestine, avoiding postoperative complications, and mitigating high mortality rates [14,15,16,17].

Acute appendicitis (AA) is one of the most frequent causes of acute abdominal pain and is one of the main indications for urgent surgery worldwide; according to this, appendectomy is a high-volume procedure, and the laparoscopic approach (laparoscopic appendectomy, LA) has shown advantages in terms of clinical results and cost-effectiveness, above all if the endo-loop is used as the device chosen to ligate the appendiceal stump [18,19].

According to the above-mentioned background, we thought of testing intraoperatively a new hypothetical challenging use of ICG-FA for this very widespread emergency surgical procedure, assured by this dye’s safety in colorectal surgery. 

As far as we know, there are no published manuscripts or case reports regarding this topic or this new possible use of ICG-FA in the literature.

Our main aims were to evaluate the appendicular base before proceeding with resection and the appendicular stump, to decide whether the use of an endo-stapler could be secondly suggested.

Our patients had no immediate complications after the surgical procedure, and the 30-day follow-up was regular, without wound infections, intra-abdominal post-operative abscesses, or other complications according to the Clavien-Dindo classification [20]. 

We are aware that our experiment has many limitations, such as the limited number of patients involved and the absence of standardization, but ICG fluorescence could be a useful method to assess stump vascularization, guide the following surgical strategy, and eventually reduce postoperative complications. 

We performed a search on PubMed using the following key words: acute appendicitis, laparoscopic appendectomy, indocyanine-green, and fluorescence angiography. To the best of our knowledge, no manuscripts concerning the use of ICG-FA to evaluate vascularization, the trophism of the appendicular basis, or the stump were found regarding this specific application.

We are aware that a randomized pilot study, also considering cost-effectiveness, is mandatory to explore this topic more deeply. 

## 5. Conclusions

Our experiment, even if limited, could suggest that ICG-FA could be one of the decision-making modalities for patients with complicated acute appendicitis to manage the appendicular base and stump with the different devices, reducing the use of endo-staplers in well-selected patients. 

We are aware that further studies are necessary.

In our mini-series, no patients developed postoperative complications such as IAA, wound infections, or others. The postoperative course was uneventful. *P1* and *P2* were discharged on post-operative day (POD) two, while *P3*, *P4*, and *P5* were discharged on POD three. A 30-day follow-up was performed with these patients without finding any adverse effects.

Larger, further randomized prospective trials are needed to standardize, test, and eventually validate this new technique.

## 6. Key Messages

ICG-FA has feasibility and usefulness in the intraoperative assessment of colorectal and intestinal surgical procedures.ICG-FA could be a useful method for assessing stump vascularization, guiding the following surgical strategy, and eventually reducing postoperative complications.As far as we know, there are no published manuscripts or case reports regarding this topic in the literature or this new possible use of ICG-FA, and larger, further randomized trials are needed.

## Figures and Tables

**Figure 1 jcm-12-05173-f001:**
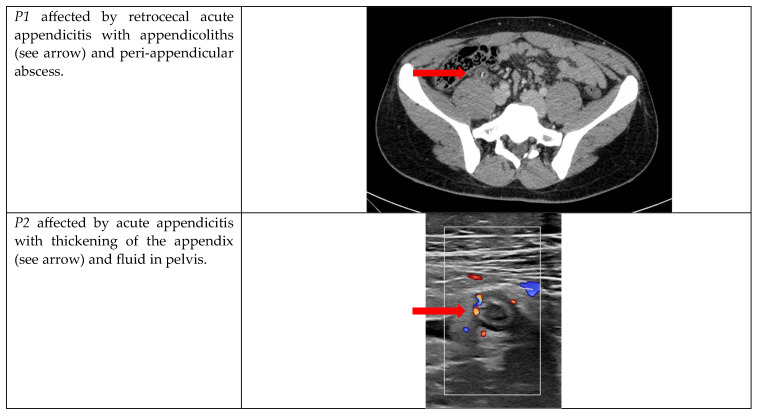
Radiological features.

**Figure 2 jcm-12-05173-f002:**
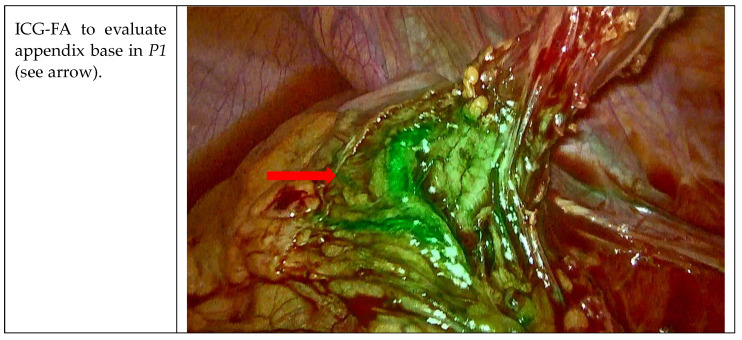
Visualization of the appendicular bases with ICG-FA.

**Figure 3 jcm-12-05173-f003:**
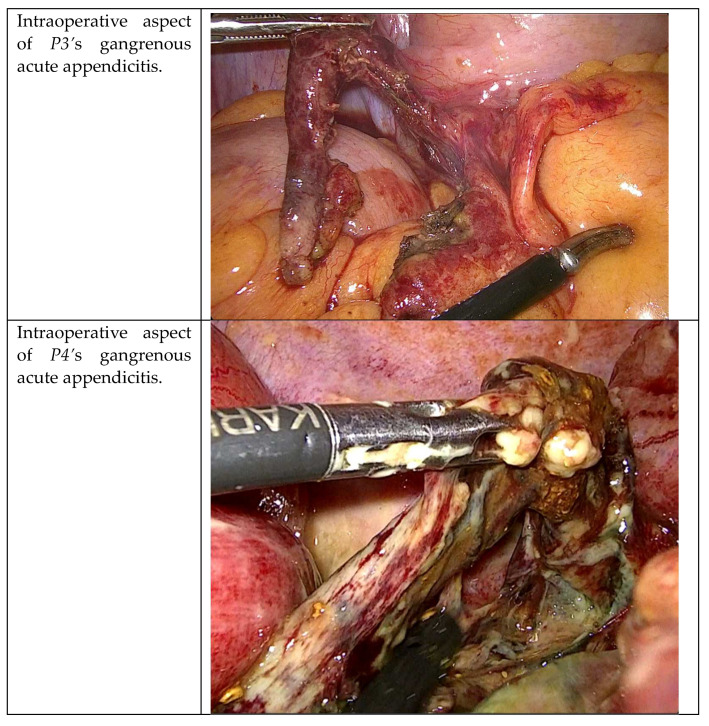
Intraoperative WL presentation of gangrenous appendicitis in *P3*, *P4*, and *P5*.

**Figure 4 jcm-12-05173-f004:**
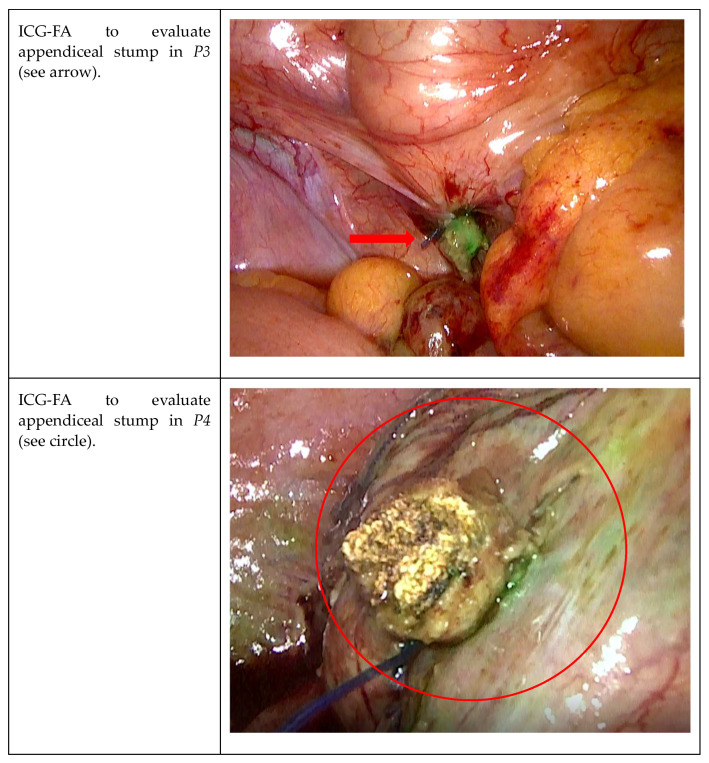
Visualization of the appendicular stump with ICG-FA.

**Table 1 jcm-12-05173-t001:** Socio-demographic characteristics.

Patient	Sex	Age	BMI	Laboratory Tests	Comorbidities	Etnicity
*P1*	M	20 years	24.7	WBC 17.14 10^9^/L, CRP 0.08 mg/dL	No	Caucasian
*P2*	F	18 years	29.8	WBC 19.55 10^9^/L, CRP 3.52 mg/dL	No	Caucasian
*P3*	M	79 years	27.5	WBC 12.23 10^9^/L, CRP 1.79 mg/dL	Atrial fibrillation in treatment with new oral anticoagulant; high blood pressure	Caucasian
*P4*	M	52 years	26.8	WBC 22.37 10^9^/L, CRP 17.19 mg/dL	No	Caucasian
*P5*	M	65 years	27.9	WBC 18.06 10^9^/L, CRP 7.48 mg/dL	No	Caucasian

## Data Availability

At the moment there is not this possibility.

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
