# Peer review of "A Hypothetical New Challenging Use for Indocyanine Green Fluorescence during Laparoscopic Appendectomy: A Mini-Series of Our Experience and Literary Review"

_jcm, 2023, doi:10.3390/jcm12165173_

Round 1

Reviewer 1 Report

At first glance, the manuscript does not have the structure of an original scientific article, which must consist of an introduction, materials and methods, a discussion and a conclusion. In relation to the content of the manuscript, I suggest you consider the "Technical note" category.

Before listing the cases themselves, it is unclear what hypothesis and aims the authors set for themselves!?

When enumerating the cases themselves, the impression of monotony is left. Readers must already have an impression from the description of the cases, what is the focus of the manuscript. What did you want to achieve with this method in acute appendicitis? Is the method justified in relation to the price and the current sufficient diagnostic and therapeutic procedures?

If you aimed for a clearer visualization of the appendix stump, then you must focus the entire manuscript on the visualization of the appendix stump and its possible complication - stump appendicitis.

In relation to existing methods, new equipment and excellent camera resolutions, you must have strong evidence why this method would have additional benefits in specific indications, which was missed in your manuscript.

The conclusion is completely missed. Certainly, the complications will not solely depend on the method used, but on a number of parameters.

The English language needs moderate corrections.

Reviewer 2 Report

I miss an introductory paragraph on acute appendicitis, its high incidence (and therefore its high social and health care costs) and the need to optimize its diagnosis and management. I recommend emphasizing the new diagnostic lines aimed at preoperative discrimination between complicated and uncomplicated appendicitis, which could justify the preoperative decision to prepare indocyanine green for surgical use (since its usefulness is probably higher and more justified in complicated acute appendicitis). It would also be interesting to comment that although this has not yet been applied to pediatric populations this is a promising field. I attach illustrative references

Dale L. The Use of Procalcitonin in the Diagnosis of Acute Appendicitis: A Systematic Review. Cureus. 2022 Oct 14;14(10):e30292. doi: 10.7759/cureus.30292. PMID: 36407148; PMCID: PMC9655768.

Turkes GF, Unsal A, Bulus H. Predictive value of immature granulocyte in the diagnosis of acute complicated appendicitis. PLoS One. 2022 Dec 21;17(12):e0279316. doi: 10.1371/journal.pone.0279316. PMID: 36542634; PMCID: PMC9770334.

Arredondo Montero J, Antona G, Rivero Marcotegui A, Bardají Pascual C, Bronte Anaut M, Ros Briones R, Fernández-Celis A, López-Andrés N, Martín-Calvo N. Discriminatory capacity of serum interleukin-6 between complicated and uncomplicated acute appendicitis in children: a prospective validation study. World J Pediatr. 2022 Dec;18(12):810-817. doi: 10.1007/s12519-022-00598-2. Epub 2022 Sep 16. PMID: 36114365; PMCID: PMC9617836.

Arredondo Montero J, Rico Jiménez M, Martín-Calvo N. Discriminatory capacity of serum total bilirubin between complicated and uncomplicated acute appendicitis in children: a systematic review and a diagnostic test accuracy meta-analysis. Pediatr Surg Int. 2022 Dec 27;39(1):64. doi: 10.1007/s00383-022-05352-3. PMID: 36574051.

The manuscript needs an English revision by a native speaker. There are typos and expressions that sound strange and unnatural (“to manage the base of the appendiceal…”)(“ cutten”)(“ recente literature”)

Mini-series and evolution:

-          Eliminate the hospital. Not relevant

-          Eliminate that the patient was autonomous in his daily activities. He is 20 years old, obviously unless he had serious associated pathology this is normal. The same for patient 2.

-          Eliminate the normality ranges; these are common markers and the ranges are known to any surgeon.

-          The radiological images are of very low quality. They look like a photograph of the physical slide put in a negatoscope. I recommend digitizing them, if so, to improve the quality (if possible). The mouse arrow on the ultrasound image is unacceptable.

-          Do not start every new patient with “in addition” or with “moreover”. Start as a new paragraph “…A 18-year-old woman…”

-          It strikes me that authors used a 10 mm trocar on the flank, when the usual procedure is that the accessories are 5 mm. Why?

-          “All patients were adult, had no allergy to Iodine and were not in therapy for hyperthyroidism.”. This has already been expressed in the description of each patient and is redundant (except for hyperthyroidism). Try to be succinct and efficient in the wording. Same with informed consent: reference is made at two points to obtaining informed consent; unify please.

-          “tissues_ perfusión”. Typo?

-          “half dose of 25 mg”. This expression is confusing.

-           Merge radiological figures in a collage and surgical figures in a collage. Simplify the layout and visualization of the article.

-          ICG-FA images are highly variable, some good and some very bad. The authors do not discuss the time from administration to photography or the technical characteristics of the laparoscopy tower.

-          Surgical images would benefit from the use of arrows identifying the structures.

-          A table with the clinical and socio-demographic characteristics of the patients should be generated.

-          The costs and increase in surgical time associated with this technical innovation should be discussed.

-          The authors should specify whether this work has been approved by the local ethics committee and provide the registration code.

The manuscript needs an English revision by a native speaker. There are typos and expressions that sound strange and unnatural (“to manage the base of the appendiceal…”)(“ cutten”)(“ recente literature”)

Round 2

Reviewer 1 Report

I appreciate your efforts to improve the manuscript.

Moderate editing of English language required.

Reviewer 2 Report

Overall the improvement of the manuscript is remarkable. Some additional corrections:

1.       I find acceptable what the authors raise in relation to the introduction and the pediatric population. I do not need any further changes in this regard. However, I believe that the authors should be a little more cautious in their introduction: at present, non-surgical management of acute appendicitis has acceptable initial success rates but a high rate of therapeutic failure in the first 12 months after the episode, so it is not the gold standard. Likewise, gangrenous appendicitis, without perforation, is considered subsidiary to having suffered bacterial translocation and in general terms the literature considers it a complicated appendicitis.

2.       Avoid the expression "an idea came up". It is not academic

3.       Do not use abbreviations not previously clarified (ED).

4.       Check the manuscript carefully, and try to correct any typos "appendicolitis".

5.       I appreciate the addition of the sociodemographics table, but the authors have to try to find the relevant data. Marital status and occupation are irrelevant.

6.       Table 2 is more of a figure than a table. Same for 3,4,5

7.       The authors did a "search" in pubmed, not a "research" (same for line 187). The authors should not conclude so categorically that there are no publications on this topic. They have not checked other bibliographic databases (Scopus, WoS, Embase, Scielo...) and have not performed Boolean searches. I recommend to qualify (“to the best of our knowledge, we did not find previous publications regarding this specific application for ICG…”

8.       I think it would also be interesting to include arrows and signs in the surgical images. However, if the authors believe that this would make the visualization of the images more difficult, I understand their omission.
